# MicroRNA-769-3p Acts as a Prognostic Factor in Oral Squamous Cell Cancer by Modulating Stromal Genes

**DOI:** 10.3390/cancers14184373

**Published:** 2022-09-08

**Authors:** Heejin Lee, Sang Hoon Chun, Seo Yun Moon, Jung-Sook Yoon, Hye Sung Won, Soon Auck Hong, Seo Ree Kim, Kwang-Jae Cho, Keunsoo Kang, Sieun Lee, Young-Ho Ahn, Ji Hyung Hong, Yoon Ho Ko

**Affiliations:** 1Department of Internal Medicine, College of Medicine, The Catholic University of Korea, Seoul 06591, Korea; 2Cancer Research Institute, College of Medicine, The Catholic University of Korea, Seoul 06591, Korea; 3Division of Oncology, Department of Internal Medicine, College of Medicine, The Catholic University of Korea, Seoul 06591, Korea; 4Uijeongbu St. Mary’s Hospital Clinical Research Laboratory, The Catholic University of Korea, Uijeongbu-si 11765, Korea; 5Department of Pathology, College of Medicine, Chung-Ang University, Seoul 06973, Korea; 6Department of Otolaryngology-Head and Neck Surgery, The Catholic University of Korea College of Medicine, Seoul 06591, Korea; 7Department of Microbiology, College of Science & Technology, Dankook University, Cheonan 31116, Korea; 8Department of Molecular Medicine, College of Medicine, Ewha Womans University, Seoul 07804, Korea

**Keywords:** miR-769, head and neck neoplasm, stroma, tumor-suppressive, EMT

## Abstract

**Simple Summary:**

Based on a recent report, miRNA expression profiles represent the head and neck squamous cell carcinoma (HNSCC) subtype (epithelial or stromal), owing to the ability of miRNAs to regulate developmental growth and differentiation programs in epithelial cells. The current study aimed to investigate the expression and function of miR-769-3p in the stromal phenotype of oral squamous cell cancer (OSCC). Our results showed that miR-769-3p is overexpressed in fibroblast-like cells within the stroma around tumor tissues. miR-769-3p overexpression inhibited the proliferation, migration, and invasion of OSCC cells. RNA sequencing revealed a tumor-suppressive role of miR-769-3p by modulating stromal gene expression, such as interferon-gamma-related genes and MYC target gene sets. miR-769-3p may be a potential biomarker of the miRNA phenotype in OSCC patients. These findings indicate that miR-769-3p plays a crucial role as a potential biomarker of the miRNA phenotype in OSCC patients.

**Abstract:**

miR-769-3p expression is suppressed in the stromal subtype of head and neck squamous cell carcinoma (HNSCC); however, its role in stromal HNSCC has not been fully elucidated. To investigate the biological relevance of miR-769-3p in the stromal phenotype, we established oral squamous cell cancer (OSCC) cell lines, namely CAL27, HSC3, and YD8, overexpressing miR-769-3p. miR-769-3p expression was positively and negatively correlated with interferon-gamma-related genes and MYC target gene sets, respectively. miR-769-3p decreased OSCC cell migration and invasion as well as mesenchymal marker expression and increased epithelial marker expression. Moreover, miR-769-3p enhanced OSCC cell sensitivity to 5-fluorouracil. High miR-769-3p expression was associated with good prognosis of HNSCC patients. Collectively, these results suggest that miR-769-3p suppression enhances stromal gene expression and promotes the epithelial-to-mesenchymal transition. Therefore, miR-769-3p may be a potential biomarker of the miRNA phenotype in OSCC patients.

## 1. Introduction

Globally, head and neck squamous cell carcinomas (HNSCCs), including oral squamous cell carcinoma (OSCC), constitute the sixth most common cancers with a yearly world incidence of approximately 600,000 [1]. HNSCC has early recurrence or metastasis and a high mortality rate even after curative treatment. Despite the development of novel treatments, including targeted therapies and immunotherapies, the prognosis of recurrent or metastatic HNSCC is mostly poor owing to the high prevalence of local progression and distant metastasis [2,3]; therefore, a new therapeutic strategy is needed to prevent recurrence and metastasis. The metastatic potential of cancer cells is determined by their interactions with the surrounding stromal cells, such as fibroblasts, immune cells, and vascular endothelial cells that compose the tumor microenvironment, and is often accelerated by the epithelial–mesenchymal transition (EMT) in HNSCC.

MicroRNAs (miRNAs) are small, single-stranded, non-coding RNAs that regulate gene expression post-transcriptionally [4]. miRNAs negatively regulate target mRNAs by binding to their 3′-untranslated region (3′-UTR) to induce their degradation and interfere in the translation process. The dysregulation of the expression of various miRNAs contributes to tumorigenesis, including abnormal transcriptional regulation and epigenetic modifications of specific tumor suppressor genes or oncogenes. Increasing evidence shows that the dysregulation of certain miRNAs, such as miR-21, miR-26a/b, and miR-137, is related to HNSCC development and progression [5,6,7,8].

Moreover, based on a recent report, miRNA expression profiles represent pathologically well-known altered oncogenic and tumor-suppressive pathways and mutations in squamous tumors as well as the HNSCC subtype (epithelial or stromal), owing to the ability of miRNAs to regulate developmental growth and differentiation programs in epithelial cells [9]. These patterns suggest an axis of cancer differentiation along a stromal–epithelial spectrum influenced by miRNAs and support the need of a miRNA-based classification system in HNSCC. A total of 42 miRNAs, including miR-17, miR-200 families, and miR-769-3p, were found to be downregulated in miRNA-based stromal phenotype; this finding suggests that they represent a poorly differentiated, stromal-like pan-cancer subtype. Overall, 8 miRNAs, including miRs-150, miR-127, and miR-493 were downregulated in epithelial phenotype. These subtype-specific miRNAs regulate similar biological processes involved in the pathogenesis and prognosis of HPV-infected tumors of epithelial cells [9].

In addition, of the two miRNA-based HNSCC subtypes, the stromal subtype comprises a higher proportion of oral cavity cancers, whereas the epithelial subtype includes a greater proportion of oropharyngeal, hypopharyngeal, and laryngeal tumors. miR-769-3p is upregulated in the epithelial subtype and suppressed in the stromal subtype of HNSCC; thus, miR-769-3p suppression enhances stroma-related gene expression [9]. miR-769-3p dysregulation has been reported to have a tumor-suppressive role in several types of human cancers, such as colorectal cancer, lung cancer, hepatocellular carcinoma, and glioma [10,11,12,13]. However, the exact clinical and functional roles of miR-769-3p in OSCC remain largely unknown. This study aimed to investigate the relevance of miR-769-3p in the stromal phenotype of OSCC and further explore its biological role and mechanism in OSCC.

## 2. Materials and Methods

### 2.1. Cell Culture

The OSCC lines CAL27, HSC3, and YD8, were obtained from the American Type Culture Collection (Manassas, VA, USA), JCRB cell bank (Osaka, Japan), and Korean Cell Line Bank (Seoul, Korea), respectively. CAL27 and HSC3 cells were maintained in Dulbecco’s modified Eagle’s medium (HyClone, Logan, UT, USA), and the YD8 cell line was maintained in RPMI 1640 (HyClone) containing 10% fetal bovine serum (HyClone) and 1% antibiotics at 37 °C in the presence of 5% CO_2_. miRNAs (control miRNA [Ctrl miRNA], and miR-769-3p mimic (Bioneer, Daejeon, Korea)) were transiently transfected using Lipofectamine 2000 (Invitrogen, Carlsbad, CA, USA) according to the manufacturer’s instructions. To establish a stable cell line expressing miR-769-3p, a genomic DNA region containing human pre-miR-769-3p (619 bp) was obtained through PCR from HEK293T genomic DNA and subcloned into the pLVX-Puro vector (Clontech, Mountain View, CA, USA). pLVX-Puro_miR-769-3p was introduced into the cells via lentiviral infection.

### 2.2. Quantitative Reverse Transcription-PCR (qRT-PCR)

To prepare total RNA, cells were harvested using the TRIzol reagent (Invitrogen). miRNA levels were quantified using the HB miR Multi Assay Kit™ (Heimbiotek, Seongnam, Korea) according to the manufacturer’s instructions. The abundance of transcripts was assessed by quantitative PCR (qPCR) analysis using SYBR Green PCR Master Mix (Bioline) and gene-specific primer sets on a StepOne Plus instrument (Applied Biosystems, Foster City, CA, USA). *GAPDH* mRNA and *RNU6B* RNA were used as internal controls for normalization. The primer sequences are listed in Table 1.

### 2.3. Western Blotting

Cells were lysed in RIPA buffer (Elpis Biotech, Daejeon, Korea, #EBA-1149). Cell lysates were separated through SDS-PAGE, transferred onto PVDF membranes, and then incubated with primary antibodies followed by HRP-conjugated secondary antibodies (Genetex, Irvine, CA, USA). Chemiluminescent signals were visualized using the NEW Clarity ECL substrate (GE Healthcare, Chicago, IL, USA). Antibodies against ZEB1 (Santa Cruz Biotechnology, CA, USA #sc-25388), vimentin (Santa Cruz Biotechnology, #sc-5565), E-cadherin (Santa Cruz Biotechnology, #sc-8426), cleaved PARP1 (c-PARP; Abcam, Cambridge, MA, USA, ab32064), and GAPDH (Genetex, GTX100118) were purchased from Santa Cruz Biotechnology.

### 2.4. Spheroid Invasion Assay

To create spheroids, cells (1 × 10^5^ cells/5 mL) in 20% METHOCEL™ (Sigma-Aldrich) and 1% Matrigel™ (BD Biosciences, Franklin Lakes, NJ) were hung on the lid of 15 cm dishes and incubated at 37 °C for 2 days. A spheroid mixture (spheroids in 0.5× PBS, 0.01 N NaOH, 3 mg/mL collagen, and spheroids) was then implanted in the center of each well of a 12-well plate. After the gels were polymerized, the wells were filled with cell culture media. Phase contrast microscopy was used to capture images of invading cells. The invasion ratio was calculated by dividing the total invading area by the central spheroid area measured using Image J (http://imagej.nih.gov/ij (accessed on 1 August 2019)).

### 2.5. Colony-Formation Assay

After incubating cells with 5-fluorouracil (5-FU) or DMSO for 72 h, 1 × 10^3^ cells per well were cultured for 2 weeks. Cells were fixed with 4% paraformaldehyde, stained with 0.05% crystal violet for 10 min at room temperature, and then washed and dried.

### 2.6. RNA Sequencing

Total RNA was isolated from CAL27 cells transfected with Ctrl miRNA or miR-769-3p using the AccuPrep Universal RNA Extraction Kit (Boineer, Daejeon, Korea). RNA sequencing was performed by Macrogen (Seoul, Korea). After checking the quantity and integrity of total RNA, an RNA library was constructed with 1 µg of total RNA using the Illumina TruSeq Stranded mRNA Sample Prep Kit (Illumina, San Diego, CA, USA). Paired-end (2 × 100 bp) sequencing of the indexed library was performed using Illumina NovaSeq (Illumina), and RNA sequencing data were analyzed using the Octopus-toolkit [14].

### 2.7. Gene Set Enrichment Analysis (GSEA)

GSEA was conducted using the GSEAPreranked tool in the GSEA application (version 3.0) [15]. The log2 ratio of FPKM values (miR-769-3p overexpression over wild-type) was used as an input for GSEAPreranked analysis.

### 2.8. Patient Samples and Tissue Microarray (TMA) Construction

To compare miR-769-3p expression between the tumor and adjacent normal tissue resected from patients with HNSCC, TMAs were constructed for in situ hybridization (ISH). Tissue cores with 2-mm diameters were taken from morphologically representative tissues of each tumor paraffin block using a manual TMA device (SuperBioChips Laboratories, Seoul, Korea). To minimize any sampling error and reduce the impact of tissue loss during processing, duplicate tissue cores for each specimen were arrayed on a second recipient paraffin block. The array blocks were cut into 4-μm-thick sections for ISH of miR-769-3p. Both tumor and adjacent non-cancerous tissues were obtained from the available formalin-fixed paraffin-embedded (FFPE) specimens. This study was approved by the Institutional Research Ethics Board of Bucheon St. Mary’s Hospital of the Catholic University of Korea (No. HC19SESI0076) and adhered to the tenets of the Declaration of Helsinki.

### 2.9. RNA Extraction from FFPE Tumors

Total RNA was extracted from FFPE tumors of the patients using the miRNeasy FFPE Kit (QIAGEN, Hilden, Germany) according to the manufacturer’s instructions. RNA quantity and quality were assessed using a NanoDrop 2000 (Thermo Fisher Scientific, Waltham, MA, USA).

### 2.10. ISH for miR-769-3p

ISH was performed using miRCURY LNA miRNA ISH buffer and a control kit (Exiqon, Vedbaek, Denmark). Paraffin sections from each tumor were deparaffinized and predigested with proteinase K (20 μg/mL) at 37 °C for 15 min. The sections were then incubated with 100 nM double digoxigenin (DIG)-labeled LNA hsa-miR-769-3p probe (Exiqon, sequence: 5′-AACCAAGACCCCGGAGAT-3′) for 19 h at 40 °C in a StatSpinThermoBrite Slide Hybridizer (Fisher Scientific, Westwood, MA). LNA U6 snRNA positive control probe and LNA scramble miRNA negative control probe (Exiqon) were used in each experiment. After stringent washing in SSC buffer, sections were incubated with blocking solution and then incubated with anti-DIG–alkaline phosphatase (AP) antibody (1:100, Roche, Mannheim, Germany) for 1 h at room temperature. For colorimetric detection of AP, sections were incubated with NBT/BCIP (Roche) for 2 h at 30 °C. The AP reaction was stopped using KTBT buffer, and the sections were counterstained with nuclear fast red.

The intensity of miR-769-3p expression was graded as follows: 0, no staining; 1, mild tumor cell staining; 2, moderate tumor cell staining; and 3, strong tumor cell staining (Appendix A). The results were analyzed by a single pathologist (S.A.H.) who was blinded to the clinical data.

### 2.11. Data Acquisition and Microarray Data Analysis

To investigate the prognostic role of miR-769-3p in HNSCC, an available miRNA expression microarray dataset (GSE144711) from 115 patients with HNSCC was obtained from the Gene Expression Omnibus (GEO) database. A total of 53 tumor arrays remained after the removal of low-quality HNSCC samples and non-OSCC, and these were used for survival analysis.

### 2.12. Statistical Analysis

Data are presented as mean ± standard deviation (SD). The data were analyzed with Student’s t-test using GraphPad Prism (La Jolla, CA, USA). The chi-square test was performed to confirm the correlation between clinical features and miR-769-3p expression. The Kaplan–Meier method and log-rank test were used for survival analysis.

## 3. Results

### 3.1. miR-769-3p Is Overexpressed in HNSCC, Especially in Fibroblast-like Cells within the Stroma around Tumor Tissues

Prior to elucidating the biological effect of miR-769-3p on tumor progression in HNSCC, miR-769-3p expression was evaluated using GENT2 in a variety of carcinomas. Among 19 carcinomas, 10 carcinomas, including bladder cancer, breast cancer, and HNSCC, showed increased miR-769-3p expression compared to healthy controls (Figure 1A). To investigate miR-769-3p expression in HNSCC, miR-769-3p levels were compared between matched tumor and adjacent normal tissues in 20 patients with oral cavity cancer; miR-769-3p levels were significantly increased in the tumor tissues compared to the normal adjacent tissues (*p* = 0.033) (Figure 1B). To further examine the exact localization of miR-769-3p in tumor tissues, we performed miR-769-3p ISH using tumor tissues. miR-769-3p was highly expressed predominantly in fibroblast-like cells within the stroma surrounding the tumor nests but not in the intra-tumoral lesion (Figure 1C,D). The intensity of miR-769-3p ISH was higher in the stroma surrounding the tumor than in the tumor tissue itself (*p* = 0.014) (Figure 1D).

### 3.2. RNA Sequencing Reveals a Tumor-Suppressive Role of miR-769-3p by Modulating Stromal Gene Expression

We evaluated the expression levels of miR-769-3p in three different OSCC lines, of which miR-769-3p expression in CAL27 cells was the lowest (Figure 2A). To analyze the change in the transcriptome profile influenced by miR-769-3p expression, after the transfection of miR-769-3p in CAL27 cells (Figure 2B), we performed RNA-sequencing analysis using three different batches of miR-769-3p-transfected CAL27 cells. Figure 2C shows the significantly upregulated (n = 2694) and downregulated genes (n = 2312; fold-change ≥ 2 and *p* ≤ 0.05) in miR-769-3p-transfected cells. To gain insight into the biological functions of miR-769-3p, we performed an Ingenuity Pathway Analysis (IPA) (Figure 2D). The canonical mTOR signaling pathway was most significantly enriched in miR-769-3p (*p* = 5.70 × 10^−4^). To further analyze the signalling pathways involved in the development of the molecular phenotypes affected by miR-769-3p, we performed GSEA and identified enriched gene signatures in the RNA-sequencing profiles. Gene signatures including interferon-gamma-related genes (normalized enrichment score, NES = 2.45) were enriched, whereas MYC target gene signatures (NES = −4.82) were depleted in miR-769-3p-overexpressing cells (Figure 2E). qRT-PCR analyses also confirmed that immune-related genes were upregulated, whereas MYC-related genes were downregulated in miR-769-3p-overexpressing cells (Figure 2F). These findings suggest that miR-769-3p acts as a modulator of stromal gene expression.

### 3.3. Transduction of miR-769 Decreases EMT and Cancer Cell Invasion

After gaining the inference of miR-769-3p as a tumor-suppressive miRNA in oral cancer cell lines, we investigated the changes in EMT-related proteins following miR-769-3p transfection. We established miR-769-overexpressing cells in three different oral cancer cell lines through lentiviral transduction (Figure 3A). Western blotting analysis showed that miR-769-3p decreased the expression of mesenchymal markers, ZEB1 and vimentin, and enhanced the expression of the epithelial marker E-cadherin (Figure 3B), indicating that the ectopic expression of miR-769-3p reduced EMT activity. Additionally, in spheroid invasion assays, miR-769-3p suppressed the invasion of HSC3, CAL27, and YD8 cancer cell spheroids into collagen gels (*p* < 0.001, Figure 3C). These results suggest that miR-769-3p reduces EMT and the invasion of OSCC cells.

### 3.4. miR-769-3p Overexpression Results in the Increased Sensitivity of OSCC to 5-Fluorouracil Chemotherapy

Since EMT-related signaling pathways are involved in drug sensitivity, we sought to determine the effect of miR-769-3p on chemotherapy sensitivity in OSCC. CAL27 cells were transfected with miR-769-3p or Ctrl miRNA, and then treated with 5-FU for 72 h. Ectopic miR-769-3p significantly decreased cell viability and increased c-PARP protein levels, as an indicator of apoptosis, after 5-FU treatment (Figure 4A,B). We also observed a decrease in the numbers in miR-769-3p-overexpressing cells after 5-FU treatment (Figure 4C).

### 3.5. High miR-769-3p Expression Is Correlated with Longer Survival in HNSCC Patients

We assessed the correlation of miR-769-3p expression levels with clinical prognosis through Kaplan–Meier curve analysis with a log-rank test using the miRNA expression microarray dataset (GSE144711, OSCC, n = 53). A high miR-769-3p expression tended to result in longer overall survival (*p* = 0.094, Figure 5A, Appendix A). In addition, we compared the clinical role between tumor and stroma cells miR-769-3p expression using ISH in the OSCC FFPE samples (n = 31). Patients with high miR-769-3p expression in tumor tissue showed longer relapse-free survival (*p* = 0.042, Figure 5B) and overall survival (*p* = 0.095, Figure 5C). However, the expression of miR-769-3p expression in stroma showed no significant correlations with clinical outcomes (Appendix A).

## 4. Discussion

Epithelial tumor cells acquire mobility and invasiveness through the EMT process, especially from communication with the tumor microenvironment, which is controlled by various transcription factors and miRNAs [2]. EMT predominantly determines the metastatic potential of cancer cells through their interactions with the surrounding stromal cells in the tumor microenvironment. OSCC predominantly includes the mesenchymal molecular subtype [16] or the miRNA-based stromal subtype [9]. Our findings indicate that miR-769-3p has tumor-suppressive mechanisms from its modulation of stromal gene signatures, including MYC-related genes or inflammation gene signatures, in OSCC. We also confirmed that miR-769-3p enhanced the chemosensitivity of OSCC cells to 5-FU chemotherapy. In addition, there was a tendency for longer survival duration of HNSCC patients with higher miR-769-3p expression than those with lower expression.

Recently, robust miRNA clusters have been used to subclassify HNSCC tumors, as well as other squamous tumors of other anatomic locations. miRNA-based profiles can distinguish between epithelial and stromal tumor subtypes. In particular, the prognostic value of epithelial and stromal miRNA-based subtypes showed more significant differences in survival between subtypes than mRNA-based subtypes [9,16]. The difference in prognosis may be explained by multiple mechanisms, such as increased EMT, cell motility, proliferation, and angiogenesis in the stromal subtype. The miR-200 family along with miR-205-5p and miR-27b-5p serve as potent regulators of epithelial and stromal plasticity in HNSCC [9]. In addition, miR-769-3p expression is upregulated in the epithelial subtype and suppressed in the stromal subtype of HNSCC, indicating that the suppression of miR-769-3p enhances stroma-related gene expression [9]. These findings are consistent with our results. Collectively, we hypothesize that miR-769-3p is a potential biomarker of the miRNA-based stromal phenotype in patients with OSCC.

Although miR-769-3p/5p has been reported to have a conflicting biological role in various solid cancers, miR-769-3p plays a tumor-suppressive role [10,11,12,13,14,15,16,17]. In this study, we did not investigate the target mRNA of miR-769-3p; nevertheless, several target genes of miR-769-3p have been reported, and their functions in cancer have been revealed. Interestingly, these targets have been found to play key roles in EMT or tumor progression. Moreover, miR-769-3p inhibits tumor progression in glioma by suppressing ZEB2 and inhibiting the Wnt/β-catenin signaling pathway [10]. miR-769-3p suppressed the proliferation and invasion of colorectal cancer cells through Hairy/enhancer-of-split related with YRPW motif protein 1 or by directly targeting cyclin-dependent kinase 1 [11,17]. In contrast, the mutant p53–exosomal miR-769-3p–fibroblast–cytokine circuit appears to be responsible for communication between tumor and stromal cells, with exosomal miRNAs acting as a bridge, suggesting its potential role in pulmonary metastasis of colon cancer [18]. The expression level of miR-769-3p and its function in HNSCC have not yet been reported. miR-769-5p expression is downregulated in tissues or blood of patients with HNSCC [19] and lung cancer [13].

Of the 42 miRNAs downregulated in the stromal subtype of HNSCC [9], the miRNAs of the miR-200 family and miR-106b show downregulated expression in cancer stromal tissue compared with colorectal cancer epithelial tissues [20,21]. However, in this study, we observed that miR-769-3p was more downregulated in OSCC epithelial tissues than in stromal tissues. This finding suggests that the function of stromal phenotype-related miRNAs may not simply be determined by the miRNA expression site.

Coordinated oncogenic pathways are involved in the development of miRNA-based HNSCC phenotypes. We observed that MYC-related gene signatures were downregulated in miR-769-3p-overexpressing cells. Interestingly, a previous study also reported that the MYC pathway gene set was enriched in the stromal miRNA phenotype of HNSCC and correlated with miR-769-3p expression [9]. MYC is undoubtedly the master regulator of the tumor microenvironment. A sufficient c-MYC expression is essential for high proliferation rates as well as the maintenance of an undifferentiated state of human mesenchymal stromal cells and presents an increased risk of neoplastic transformation [22]. miR-769-3p downregulates n-MYC downstream-regulated gene 1 expression and enhances apoptosis in breast cancer MCF-7 cells upon changes in oxygen concentration [23].

In this study, immune-related gene signatures, including interferon-gamma signaling pathway, were enriched in miR-769-3p-overexpressing cells. RNA-sequencing analysis of miR-769-3p-overexpressing OSCC cells showed that expression of the key genes related to the interleukin-2 signaling pathway, such as IL7R, IL73RA1, and ITIH4, was strongly upregulated or downregulated, with statistical significance; this finding may confer favorable clinical outcomes in patients with HNSCC. Along these lines, in a study of gastric cancer, STAT3 binds to the miR-769 promoter, and miR-769-5p/miR-769-3p expression is negatively regulated by the STAT3–IGF1R–HDAC3 complex [24]. In addition, miR-769-5p inhibits cancer cell progression in OSCC by directly targeting the JAK1/STAT3 pathway [25].

Furthermore, our results showed that the overexpression of miR-769-3p decreased the expression of the mesenchymal markers ZEB1 and vimentin, as well as the migration and invasion of OSCC cells. In addition, miR-769-3p enhanced the chemosensitivity of OSCC cells to the anticancer drug 5-FU. We found a better survival (relapse-free survival) with higher expression of miR-769-3p in 53 HNSCC patients in the GSE dataset. This finding was reported in a previous study of patients with glioma, wherein high tissue miR-769-3p expression predicted good overall survival and was identified as an independent prognostic factor [10].

## 5. Conclusions

In this study, we found that miR-769-3p suppressed stromal gene signatures. This tumor-suppressive role of miR-769-3p may be derived from immune-related gene signature activation or c-MYC oncogene pathway suppression. Therefore, miR-769-3p may be a potential biomarker of the miRNA phenotype and in patients with OSCC, suggesting its potential for usage as an OSCC prognostic marker.

## Figures and Tables

**Figure 1 cancers-14-04373-f001:**
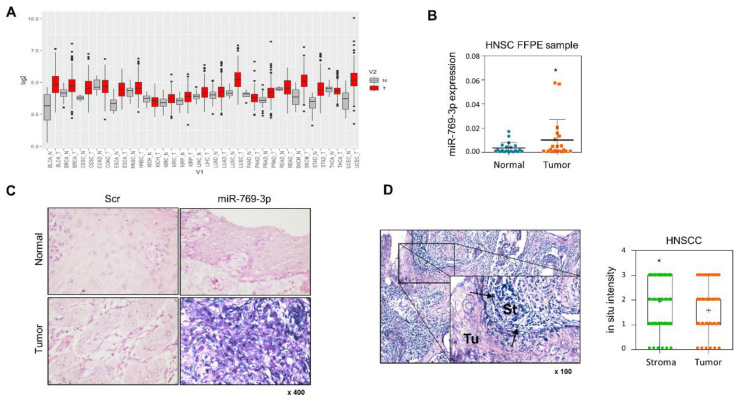
Expression pattern of miR-769-3p expression in head and neck squamous cell carcinoma (HNSCC). (**A**) The expression level of miR-769-3p in normal (gray) and tumor (red) tissues from the GENT2 data set; (**B**) qRT-PCR of miR-769-3p in tumor and normal FFPE samples from head and neck squamous cell carcinoma patients (HNSCC, n = 20); (**C**) in situ hybridization for miR-769-3p in tissue microarrays of HNSCC patients; and (**D**) localization of miR-769-3p expression in HNSCC and normal adjacent tissues using in situ hybridization. miR-769-3p was highly expressed predominantly in fibroblast-like cells (arrow) within the stroma (St) but not in the intra-tumoral lesion (Tu). Data represent mean ± SD. *P* value was determined using two-tailed Student’s *t*-test. * *p* ≤ 0.05.

**Figure 2 cancers-14-04373-f002:**
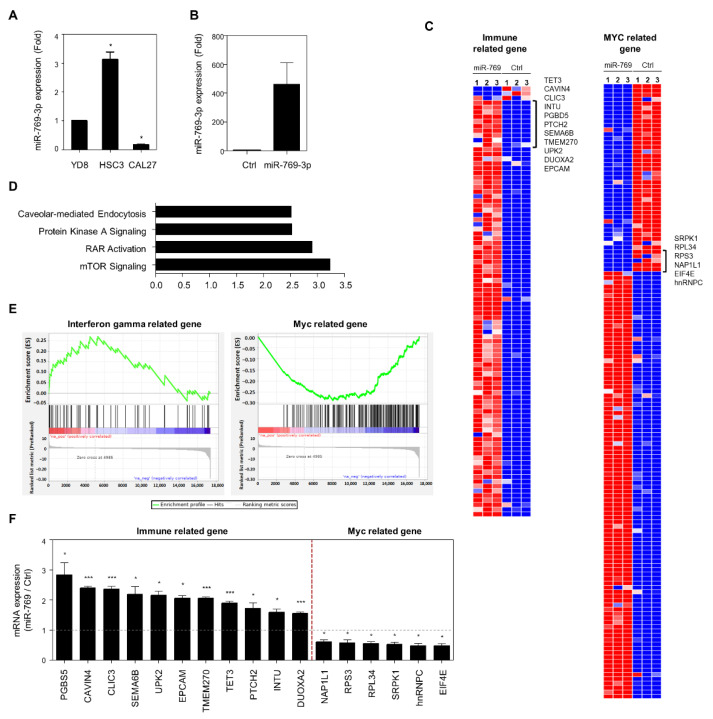
miR-769-3p overexpression promoted a stroma-related gene expression signature in oral squamous cell carcinoma (OSCC) cell lines. (**A**) Endogenous miR-769-3p expression in YD8, HSC3, and CAL27 cells; relative values to those of YD8 cells (set at 1.0) are presented; (**B**) CAL27 cells were transiently transfected with miR-769-3p or Ctrl miRNA for 48 h. miR-769-3p expression was assessed using RT-qPCR; (**C**) heatmap showing RNA-sequencing results for Ctrl miRNA- or miR-769-3p-transfected cells. Red: increased expression; blue: decreased expression; (**D**) the IPA analyses of RNA-sequencing data found canonical pathways significantly affected by miR-769-3p transfection; (**E**) gene set enrichment analysis of differentially expressed genes in Ctrl miRNA- and miR-769-3p-transfected cells. “Interferon-gamma-related” gene sets were significantly enriched, whereas the “MYC related” gene set was depleted in miR-769-3p-transfected cells; (**F**) immune-related and MYC related gene sets were validated using qRT-PCR. Relative values to those of Ctrl miRNA (set at 1.0) are presented. Data represent mean ± SD. *p* value was determined using two-tailed Student’s t-test. * *p* ≤ 0.05, *** *p* ≤ 0.001.

**Figure 3 cancers-14-04373-f003:**
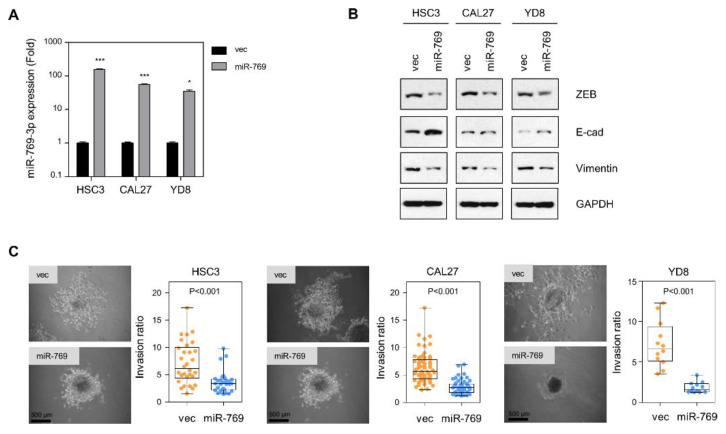
miR-769-3p overexpression decreases EMT protein levels and OSCC cell invasion. (**A**) qRT-PCR of miR-769-3p from CAL27, YD8, and HSC3 cells transduced with miR-769 or empty vectors (vec). The expression levels were normalized to the RNU6B snoRNA level; (**B**) Western blot of ZEB1, E-cadherin, and vimentin from miR-769-3p over-expressing cells. GAPDH was used as loading control. The uncropped blots are shown in Appendix A; (**C**) spheroid invasion assay using HSC3_vec and HSC3_miR-769-3p cells, CAL27_vec and CAL27_miR-769-3p cells, and YD8_vec and YD8_miR-769-3p cells. Spheroids made from hanging-drop cultures were seeded on collagen gels and then cultured. Spheroid invasion ratios (ratio of whole cell area to central spheroid area) measured using ImageJ. Data represent mean ± SD. *P* value was determined using two-tailed Student’s *t*-test. * *p* ≤ 0.05, *** *p* ≤ 0.001.

**Figure 4 cancers-14-04373-f004:**
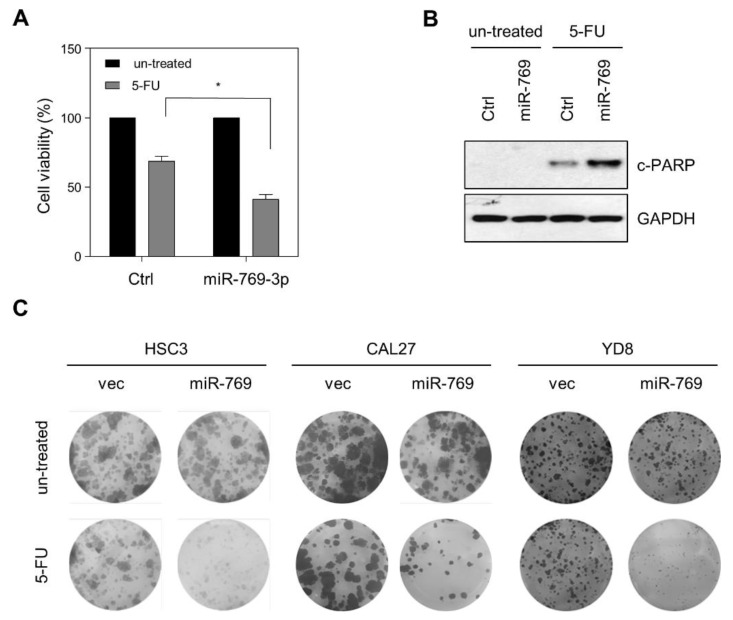
Ectopic expression of miR-769-3p alleviates the pro-survival effect in response to 5-fluorouracil (5-FU) in OSCC cell lines. (**A**) CAL27 cells were transfected with miR-769-3p and further incubated with 5-FU for 72 h. Cell viability was assessed using MTT assay; (**B**) CAL27 cells were transfected with miR-769-3p and further incubated with 5-FU for 48 h. Cleaved PARP (c-PARP) levels were assessed using Western blotting. GAPDH was used as loading control. The uncropped blots are shown in Appendix A; (**C**) to analyze the colony-forming activity of miR-769 or empty vector transduced cells lines, 1 × 10^3^ cells of each cell line were cultured for 2 weeks, and the colonies were stained with crystal violet. Data represent mean ± SD. *p* value was determined using two-tailed Student’s t-test. * *p* ≤ 0.001.

**Figure 5 cancers-14-04373-f005:**
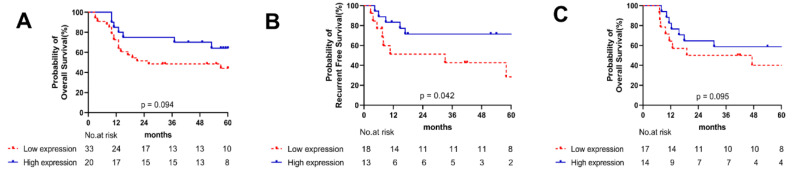
Kaplan–Meier survival curves for relapse-free survival (RFS) and overall survival (OS) based on miR-769-3p expression. High miR-769-3p expression is associated with longer OS (**A**) in HNSCC data set (GSE144711, n = 53). Patients with a high miR-769-3p expression in tumor cells demonstrated a longer RFS (**B**) and OS (**C**) than patients with a low miR-769-3p (*p* = 0.042 and *p* = 0.095, respectively) using ISH in the OSCC FFPE samples (n = 31). Patients were divided into two groups (miR-769-3p_high and miR-769-3p_low) based on their miR-769-3p expression levels. *p* values were determined using the log-rank test.

**Table 1 cancers-14-04373-t001:** List of the primer sequences used for RT-PCR analysis.

Gene	Forward Primer	Reverse Primer
PGBS5	CAGCCAAGCGATTCATTCACA	CACCATGTTCTTGAGGACGTT
CAVIN4	TGAAGTCAAGCAAGAGGAAATA	CTCTTCTTGGTTCTCAGTTAGG
CLIC3	CGCCTCGTTACAGGGAGTC	GGCACCGGGTTCTTGATGA
SEMA6B	AAACGTGTGTCGGATGAAGGG	GGGTTGAAGGCGTTGGAAC
UPK2	ACTCGGCTCAGTGCATACC	GCACCGTGATGACCACCAT
EPCAM	AATCGTCAATGCCAGTGTACTT	TCTCATCGCAGTCAGGATCATAA
TMEM270	TGGAGGGTGTGTCAGAAG	TAGTCTCCACCCACCAATAC
TET3	TCCAGCAACTCCTAGAACTGAG	AGGCCGCTTGAATACTGACTG
PTCH2	TCACACCCGAAGCACTTGG	ATCATCCGCTCAATCATTCCATT
INTU	GAGGTTGAAGGTATCCAGCAGA	GTGTCCCAGTTACGTTTTCCA
DUOXA2	GTTCATAGGCGCAGAAATTGTG	AGACGGACACGGGCTGTAA
NAP1L1	AAAGCACGTCAGCTAACTGTT	TTGAGAGCATTCACTCGTCTTTT
RPS3	AGGGCAGTGTAGAGCTTTATGC	ATGAACCGCAGCACACCATAG
RPL34	GTTTGACATACCGACGTAGGC	GCACACATGGAACCACCATAG
SRPK1	GTTCGCAATTCAGACCCTAATGA	GGATGATACGGCACTTGGTATG
hnRNPC	CCCTTCTCCGTCCCCTCTAC	CCCGAGCAATAGGAGGAGGA
EIF4E	GAAACCACCCCTACTCCTAATCC	AGAGTGCCCATCTGTTCTGTA
GAPDH	TGCACCACCAACTGCTTAGC	GGCATGGACTGTGGTCATGAG

## Data Availability

The data presented in this study are available on request from the corresponding author.

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
