# Peer review of "MicroRNA-769-3p Acts as a Prognostic Factor in Oral Squamous Cell Cancer by Modulating Stromal Genes"

_cancers, 2022, doi:10.3390/cancers14184373_

Round 1

Reviewer 1 Report (Previous Reviewer 1)

The authors have answered satisfactorily to my previous comments. The article has been greatly improved and is now suitable for publication in Cancers.

Author Response

Thank you for your valuable comments.

Reviewer 2 Report (New Reviewer)

The major issue of this study is that miR-769-3p was highly expressed predominantly in fibroblast-like cells within the stroma surrounding the tumor nests but not in the intra-tumoral lesion. However, all experiments in this study to investigate the functional role of miR-769-3p were performed in OSCC lines. The experimental approach design was problematic, and the conclusions made based on those experiments were not accurate.

Based on Fig. 2, the authors cannot conclude that “miR-769-3p acts as a tumor-suppressive miRNA by modulating stromal genes”. Why dysregulated MYC target gene signatures and interferon-gamma-related genes should be considered as stromal gene dysregulation? And why the miRNA that can upregulate interferon-gamma-related genes and downregulate MYC-related genes should be considered as a tumor suppressive miRNA?

Is there any difference of EMT and cancer cell invasion capabilities between HSC3, CAL27 and YD8 cell lines? Based on Fig. 2A, HSC3 has the highest, while CAL27 shows the lowest miR-769-3p expression. If miR-769-3p could decrease EMT and cancer cell invasion, the authors need to show the based line of EMT and invasion ability of these cell lines.

For Fig. 5 and Supplementary Fig. 2 (line 308-311), it is questionable that high miR-769-3p expression in tumor tissue shows better prognosis, while stromal expression of miR-769-3p has no effect on prognosis. Again, back to Fig. 1, if miR-769-3p expression in stroma had no clinical outcomes, why the authors emphasize that miR-769-3p is predominantly expressed in stroma, but not tumor lesion

Minor:

The authors need to give a detailed introduction about the epithelial and stromal subtypes of HNSCC.

For Fig. 1C, it’s difficult to see the staining of miR-769-3p in fibroblast-like cells. Arrows needed to show that.

In line 236-237, the sentence “mTOR signaling was found to be the canonical pathway that was most significantly enriched in miR-769-3p” is confusing.

Author Response

15/Aug/2022

Cancers

Dear Editor:

On behalf of my co-authors, I deeply appreciate the opportunity to submit a revised version of our manuscript entitled “MicroRNA-769-3p acts as tumor suppressor in oral squamous cell cancer by modulating stromal genes” for consideration for publication in Cancers. The paper was coauthored by Heejin Lee, Sang Hoon Chun, Seo Yun Moon, Jung-Sook Yoon, Hye Sung Won, Soon Auck Hong, Seo Ree Kim, Kwang-Jae Cho, Keunsoo Kang, Sieun Lee, Young-Ho Ahn, and Ji Hyung Hong.

This study aimed to investigate the relevance of miR-769-3p in the stromal phenotype of oral squamous cell carcinoma and further explore its biological role and mechanism. We believe that our study makes a significant contribution to the literature because it reveals that miR-769-3p is a potential biomarker of the miRNA-based stromal phenotype in patients with oral squamous cell carcinoma.

Further, we believe that this paper will be of interest to the readership of your journal because Caners covers aspects of cancer-based disease. As such, we believe that our manuscript is well suited to publication in Cancers.

This manuscript has not been published or presented elsewhere in part or in entirety and is not under consideration by another journal. All study participants provided informed consent, and the study design was approved by the appropriate ethics review board.  We have read and understood your journal’s policies, and we believe that neither the manuscript nor the study violates any of these. There are no conflicts of interest to declare.

Please find attached to this letter a point-by-point response to the comments of our reviewers. We are thankful for their critical input, as these suggestions greatly enhanced the quality of our manuscript.

Please let me know if we can provide any further information. We trust that our manuscript is now suitable for publication, and look forward to your decision.

Thank you for your consideration. We look forward to hearing from you.

Sincerely,

Yoon Ho Ko, MD, PhD

Professor, Department of Internal Medicine,

Division of Oncology, Department of Internal Medicine, Eunpyeong St. Mary’s Hospital, College of Medicine, The Catholic University of Korea, 1021, Tongil-ro, Eunpyeong-gu, Seoul 03312, Republic of Korea

Cancer Research Institute, College of Medicine, The Catholic University of Korea, 222, Banpo-daero, Seocho-gu, Seoul 06591, Republic of Korea

Tel.: 82-2-2030-4360

Fax: 82-2-2030-4341

E-mail: koyoonho@catholic.ac.kr

Reviewer 3 Report (New Reviewer)

The submitted manusript is a clear hypothesis-based study. Authors hypothesized that miR-769-3p might be a suppressive tumormarker in HNSCC. They overexpressed this siRNA in several HNSCC tumor cells and found that it decreased OSCC cell migration and invasion and reversed EMT, moreover induced 5-FU sensitivity. 

Comments

Do the authors have results of the effect of  miR-769-3p overexpression and cisplatin response in HNSCC/OSCC cell lines?

Introduction

Differences of miRNA expression in HNSCC in relation to HPV-background were reported. Please mention this shortly in the Introduction. Please also mention if miR-769-3p dysregulation is different in HPV-positive and negative HNSCC.

Discussion

Do the authors have a hypothesis on the effect of stroma-expressed miR-769-3p on cancer cell nests? Is it active only to tumor cells nearing expressing stroma cells or exosome or any other delivery is expected?

Author Response

15/Aug/2022

Cancers

Dear Editor:

On behalf of my co-authors, I deeply appreciate the opportunity to submit a revised version of our manuscript entitled “MicroRNA-769-3p acts as tumor suppressor in oral squamous cell cancer by modulating stromal genes” for consideration for publication in Cancers. The paper was coauthored by Heejin Lee, Sang Hoon Chun, Seo Yun Moon, Jung-Sook Yoon, Hye Sung Won, Soon Auck Hong, Seo Ree Kim, Kwang-Jae Cho, Keunsoo Kang, Sieun Lee, Young-Ho Ahn, and Ji Hyung Hong.

This study aimed to investigate the relevance of miR-769-3p in the stromal phenotype of oral squamous cell carcinoma and further explore its biological role and mechanism. We believe that our study makes a significant contribution to the literature because it reveals that miR-769-3p is a potential biomarker of the miRNA-based stromal phenotype in patients with oral squamous cell carcinoma.

Further, we believe that this paper will be of interest to the readership of your journal because Caners covers aspects of cancer-based disease. As such, we believe that our manuscript is well suited to publication in Cancers.

This manuscript has not been published or presented elsewhere in part or in entirety and is not under consideration by another journal. All study participants provided informed consent, and the study design was approved by the appropriate ethics review board.  We have read and understood your journal’s policies, and we believe that neither the manuscript nor the study violates any of these. There are no conflicts of interest to declare.

Please find attached to this letter a point-by-point response to the comments of our reviewers. We are thankful for their critical input, as these suggestions greatly enhanced the quality of our manuscript.

Please let me know if we can provide any further information. We trust that our manuscript is now suitable for publication, and look forward to your decision.

Thank you for your consideration. We look forward to hearing from you.

Sincerely,

Yoon Ho Ko, MD, PhD

Professor, Department of Internal Medicine,

Division of Oncology, Department of Internal Medicine, Eunpyeong St. Mary’s Hospital, College of Medicine, The Catholic University of Korea, 1021, Tongil-ro, Eunpyeong-gu, Seoul 03312, Republic of Korea

Cancer Research Institute, College of Medicine, The Catholic University of Korea, 222, Banpo-daero, Seocho-gu, Seoul 06591, Republic of Korea

Tel.: 82-2-2030-4360

Fax: 82-2-2030-4341

E-mail: koyoonho@catholic.ac.kr

Round 2

Reviewer 2 Report (New Reviewer)

This reviewer has no further questions.

This manuscript is a resubmission of an earlier submission. The following is a list of the peer review reports and author responses from that submission.

Round 1

Reviewer 1 Report

The manuscript of Heejin Lee and collaborators described the role of MicroRNA-769-3p in Oral Squamous Cell Cancer (OSCC). Overexpression experiments in OSCC led authors to conclude that miR-769-3p acts as a tumor suppressor miRNA, modulates stromal genes and impairs EMT.

There are several concerns which must be resolved prior publication in Cancers.

First, there are several mistakes in figure legends;

- In Figure 1D, authors presented a quantification of ISH. The diagram is quite confusing. In abscises, it is indicated Normal and Tumor, but legend indicate tumor (in red corresponding to normal) and stromal (in blue corresponding to tumor).

- In Figure 5, colors are probably inverted.

Several elements are missing. Authors work on miR-769-3p, which gene is at the origin of this miR ? miR-769-5p it is also overexpressed in stromal cells ? As this miR is implicated in non small cell lung cancer and in gastric cancer, it could be interesting to exclude a potential role of miR-769-5p in this cancer (OSCC).

The conclusions of authors appear overstated. By example, authors argue that miR-769-3p is a tumor suppressor miRNA whereas this miR is not expressed in vivo in tumor cells (figure 1). It is also difficult to conclude on an eventual role of tumor suppressor or oncogenic of this miRNA. The number of samples in TMA is not indicated. It is difficult to known if the localization of miR-769-3p is always stromal or not. It is important since in figure 5, authors analyses the survival in function of expression independently of the localization. The quantification of ISH appears simplistic (0, 1, 2, 3). It could be  interesting to shown an example of each intensity to help reader. Indicate the number of samples used.

In figure 4, Please show Parp signal in western blot (not only cParp). What is the effect of miR-769-3p expression on cell viability ? It is not possible to appreciate this in figure 4 A, but in figure 4C, colony forming activity appear weaker in Cal27 and HSC3 expressing miR-769-3p.

No target of miR-769-3p implicated in phenotypes observed is found It is a serious limit of this work. It is not sure that effect of this miR is direct or not. As several target are already known (as indicated in discussion), authors could verify that in their cells, this gene are underexpressed.

Reviewer 2 Report

Author presented a study aiming to investigate the biological relevance of miR-769-3p in the stromal phenotype. Results suggested that miR-769-3p may be a potential biomarker in oral squamous cell carcinoma.

In general, the manuscript by Yoon Ho Ko group is well-written, methods and results clearly presented and results supported the conclusions.

A few points to address before publication:

-In Figure 1 : the 20 HNSCC patients’ samples originated from which HNSCC subsite? This would be important to specify.

-The quality of figure 3 is not optimal,  please check. In 3B authors claim that  miR-769-3p decreased the expression of the mesenchymal markers, ZEB1 and vimentin, and enhanced the expression of the epithelial marker E-cadherin, however bands need to be quantified and shown.

-Figure 5. Patients' clinical outcome depends on several factors. The presented analysis should therefore consider at least patients' clinical and tumor pathological characteristics as well as tumor staging and reassess the correlation of miR-769-3p expression levels with clinical prognosis.
